# Objective Measurement of Posture and Movement in Young Children Using Wearable Sensors and Customised Mathematical Approaches: A Systematic Review

**DOI:** 10.3390/s23249661

**Published:** 2023-12-06

**Authors:** Danica Hendry, Andrew L. Rohl, Charlotte Lund Rasmussen, Juliana Zabatiero, Dylan P. Cliff, Simon S. Smith, Janelle Mackenzie, Cassandra L. Pattinson, Leon Straker, Amity Campbell

**Affiliations:** 1School of Allied Health, Curtin University, Perth, WA 6102, Australia; danica.hendry@curtin.edu.au (D.H.); charlotte.rasmussen@curtin.edu.au (C.L.R.); juliana.zabatiero@curtin.edu.au (J.Z.); l.straker@curtin.edu.au (L.S.); 2ARC Centre of Excellence for the Digital Child, Brisbane, ACT 2609, Australia; andrew.rohl@curtin.edu.au (A.L.R.); dylanc@uow.edu.au (D.P.C.); simon.smith@uq.edu.au (S.S.S.); janelle.mackenzie@qut.edu.au (J.M.); c.pattinson@uq.edu.au (C.L.P.); 3School of Electrical Engineering, Computing and Mathematical Sciences, Curtin University, Perth, WA 6845, Australia; 4Early Start, School of Education, University of Wollongong, Keiraville, NSW 2522, Australia; 5Institute for Social Science Research, The University of Queensland, Brisbane, QLD 4006, Australia; 6School of Computer Science, Queensland University of Technology, Brisbane, QLD 4000, Australia

**Keywords:** posture, movement, activity tracking, children, machine learning, review

## Abstract

Given the importance of young children’s postures and movements to health and development, robust objective measures are required to provide high-quality evidence. This study aimed to systematically review the available evidence for objective measurement of young (0–5 years) children’s posture and movement using machine learning and other algorithm methods on accelerometer data. From 1663 papers, a total of 20 papers reporting on 18 studies met the inclusion criteria. Papers were quality-assessed and data extracted and synthesised on sample, postures and movements identified, sensors used, model development, and accuracy. A common limitation of studies was a poor description of their sample data, yet over half scored adequate/good on their overall study design quality assessment. There was great diversity in all aspects examined, with evidence of increasing sophistication in approaches used over time. Model accuracy varied greatly, but for a range of postures and movements, models developed on a reasonable-sized (n > 25) sample were able to achieve an accuracy of >80%. Issues related to model development are discussed and implications for future research outlined. The current evidence suggests the rapidly developing field of machine learning has clear potential to enable the collection of high-quality evidence on the postures and movements of young children.

## 1. Introduction

The first five years of a child’s life are characterised by substantial and rapid neurophysiological development. An important aspect of development is the ability to assume different postures and perform a range of movements. Infants typically develop rapidly to be able to roll from supine to prone (~3–6 months), sit (~5–8 months), and crawl (~6–11 months) [1]. Movement capacity continues to develop throughout the toddler to preschooler phases of childhood, including learning to walk, then more dynamic and challenging tasks like climbing stairs, running, and jumping. Achieving these posture and movement abilities signals healthy development [2]. Failure or substantial delay in developing these abilities hinders full participation in society [3] and may increase risks for physical and mental health issues, for example, by reducing the ability to be sufficiently physically active [4]. There is therefore much interest from health and education professionals and parents in measuring the posture and movement of young children (0–5 years of age).

The most common method of measuring the quantity of a child’s posture and movement in clinical and research settings is through subjective interview or a survey completed by a child’s caregiver. However, these methods are known to be imprecise and biased [5]. Observation methods, either directly or from video, can be very accurate [6] and are the current gold standard. However, these approaches are limited in only capturing what a child does during the short period they are being observed/videoed, and a child may modify their behaviour when they know they are under observation [7]. Additionally, observational methods have a high human resource requirement, meaning population surveillance is not practical. Objective yet low-burden methods that can measure postures and movements over longer periods of time and in a child’s natural environment are therefore desirable.

Small, wearable sensors known as accelerometers are commonly used to quantify time spent at different physical activity intensities [8]. A recent systematic review concluded that accelerometers and accompanying physical activity intensity software were feasible for all-day assessment in children and can provide a good indication of the total amount of activity and temporal patterns of activity [8]. The commercially available software often uses count-based algorithms that sum the data over pre-set time periods (e.g., 15 s), and then uses thresholds to classify this data into different intensities of movement, such as sedentary, light, moderate, or vigorous-intensity activity [9]. These algorithms were established by comparing activity counts with gold standard energy expenditure measures and have been found to be sufficiently accurate [9]. Studies utilising this technology have been pivotal in understanding the link between childhood physical activity and health. However, categorising children’s movements into energy expenditure intensity categories overlooks potentially important aspects of specific postures and movements such as prone lying, sitting, standing, walking, and running [8]. For example, current intensity-based measures typically are not able to differentiate sitting from standing, despite these postures having different health implications [8]. Parents may also understand posture- and movement-based messages (e.g., ‘tummy time’) better than intensity-based measures (e.g., ‘moderate’ intensity). Thus, detailed information regarding the postures and movements a child performs daily will help clarify links with health and development outcomes, and refine policy, interventions, and public health messaging.

Accelerometry has also been the most frequently used hardware for posture and movement tracking in children. However, a major challenge is to provide a software solution that adequately recognises specific postures and movements. Traditionally, software was developed to predict posture and movements from key features in the raw accelerometer data, using mathematical approaches such as regression-based equations and thresholds. Key features were selected based on knowledge of each posture and movement; for example, the thigh is typically horizontal during sitting but vertical during standing. While these relatively simple algorithm approaches have demonstrated activity recognition accuracy often above 80% in adults, only limited postures and movements have been targeted [10]. Research using these approaches in young children (0–5 years old) is more limited, with lower accuracies being suggested to be related to children spending more time in other postures such as kneeling and crawling [11,12].

Recently, sophisticated machine learning computational approaches have evolved and become more accessible. Machine learning is the overarching term used to define a branch of artificial intelligence and is a rapidly advancing field [13]. When applied to wearable sensor data such as accelerometers, the models are trained to learn from the data [13], rather than follow simple rules based on human-defined key features. Machine learning software algorithms have demonstrated reasonable accuracy (>80%) in the identification of various postures and movements in adults [14]. There has been particular interest in applying machine learning to identify specific postures and movements in sporting contexts [15] and daily activity monitoring for people with movement impairments [16]. Collectively, this research has shown that adult postures and movements can be identified with reasonable accuracy (>80%) in a range of different contexts [14,15,16]. Whilst several studies have focused on the accurate identification of postures and movements using accelerometry data collected on young children, this information is yet to be synthesised.

Therefore, this systematic review aimed to answer the following research questions:How has young children’s posture and movement been objectively classified and measured using accelerometry and machine learning or other non-machine learning algorithm-based methods?What is the degree of accuracy of systems developed for the measurement of young children’s posture and movement using machine learning models, or other non-machine learning algorithm-based methods applied to accelerometry data?

## 2. Materials and Methods

This review was registered with Prospero (328600) and adheres to the Preferred Reporting Items for Systematic Reviews and Meta-Analyses (PRISMA) statement for systematic reviews.

### 2.1. Search Strategy

Six online databases (Medline, Pubmed, CINAHL, SCOPUS, EMBASE, and IEEE) were systematically searched using terms related to the concepts of “accelerometer”, “postures and movements”, “pre-school aged children”, and “machine learning”. See additional file Table A1 for a sample search strategy. Searches were limited to the English language and papers published since January 2010. The initial data base searches were completed in October 2022.

### 2.2. Eligibility Criteria

To be included in this systematic review, studies needed to meet the following criteria: to be published in a peer-review journal or conference proceedings; to use data from accelerometers or inertial measurement units (IMUs) for machine learning model or algorithm development; to use data processing and model development methods inclusive of machine or deep learning algorithms and algorithm-based approaches for semi-automated or automated posture and/or movement recognition; to have developed, validated, or utilised machine learning models for the classification and measurement of posture and/or movement; and to include data on children aged 0–5 years. Studies including typically developing children or children with clinical diagnoses were included. Studies were excluded if they were protocol or review studies, and if they were studies that did not include a posture or movement, for example, studies that were focused on the output of sleep time, energy expenditure, or levels of physical activity intensity defined by thresholds or cut points (e.g., moderate-intensity exercise).

### 2.3. Screening for Relevant Studies to Include in the Review

All retrieved papers were exported into Endnote (v20) and duplicates were removed. Title and abstract screening was performed by two researchers (DH and LS) independently, with assistance from “Research Screener version 1.0”, an artificial intelligence-based software system that iteratively learns from screening decisions to reduce the need to review irrelevant papers [17]. Any researcher disagreement on the eligibility of studies was resolved through discussion, without the need for escalation to a third reviewer. Following title and abstract screening, full text screening of articles was completed independently by the same two researchers.

### 2.4. Quality Assessment of Individual Studies Included in the Review

Quality assessment for each paper was conducted by DH, LS, and ALR using the COSMIN general recommendations for study design and criterion validity subscales [18]. These were modified slightly to fit the context and purpose of the study. For the general recommendations regarding study design, items requiring a conceptual framework (#4) and describing existing evidence (#6) were removed, as they were not necessarily representative of the quality of a study in this area, and for the criterion validity items, continuous scores (#4) and dichotomous scores (#5) were merged, as studies could appropriately use either type of score. For each of the eight study design items and five criterion validity items, each paper was scored ”good”, “adequate”, “doubtful”, “inadequate”, or “not applicable” based on the information provided. The number of “good” and “adequate” ratings were summed for each paper and item.

### 2.5. Data Extraction and Synthesis of Individual Studies Included in the Review

Twelve parameters were extracted and collated by DH, LS, ALR, AC, and CLR from the full manuscripts identified for final review. The first five parameters were related to study design and included participant details, study aims, sensor information, specific postures and movements performed, and data collection methods. Participant details included the number of participants, their age range, sex, whether they were from a clinical population or typically developing, how they were recruited, and the country the study was conducted in. Sensor information included the type of wearable movement sensor, sampling rate, and sensor location on the body. The next four parameters were related to classification model development methods and included whether papers had described a machine learning approach or non-machine learning algorithm-based approach, window details, feature extraction, and the specific machine learning algorithms used. The final parameters were related to the accuracy of the classification model developed and included the gold standard used for comparison, the validation approach used, and the overall accuracy of the system. A final table compiled the accuracy of the various classification models for each of the postures and movements assessed across all studies.

## 3. Results

An outline of the search results and study exclusions is provided in Figure 1. The initial database search identified 1663 results, of which 20 papers, reporting on data from 18 studies, met the inclusion criteria. Of these, 17 papers reported on the development and evaluation of machine learning-based approaches applied to wearable movement sensor data for the recognition of specific postures and movements in young children. The remaining three papers reported on non-machine learning algorithm-based approaches for the recognition of specific postures and movements in young children [19,20,21]. Table 1, Table 2, Table 3, Table 4, Table 5 and Table 6 provide characteristics of all the reviewed studies and are discussed in the following sections. Within each table, papers are presented chronologically by year to demonstrate the evolution of methodological approaches over time.

### 3.1. Quality Assessment

Table 1 provide detail of the quality assessment using the COSMIN guideline items on study design. For the eight components of general recommendations for study design, the total number of ‘good’ or ‘average’ ratings ranged from 1/8 to 8/8. Twelve out of 20 papers scored ≥7/8. The remaining eight papers scored ≤5/8. Papers most commonly fell short in describing their sample (e.g., exclusion criteria, inclusion criteria, participant recruitment, and whether the sample was representative of the population).


sensors-23-09661-t001_Table 1Table 1Quality assessment based on general recommendations for the design of a study.Author, DateGeneral Design ItemsNumber of Good or Adequate (out of 8)
12345678910
Parkka, 2010 [21]AGA
D
AIII4Boughorbel, 2011 [22]IAD
D
IIII1Trost, 2012 [23]GGG
G
GAAG8Suzuki, 2012 [24]DAG
G
GIII 4Nam, 2013 [25]DAG
G
GIII 4Zhao, 2013 [26]AGG
A
AGIA 7Goto, 2013 [27]AGG
G
GDII 5Hagenbuchner, 2015 [28]AGG
G
GGGA 8Hegde, 2018 [29]GGG
G
AAIA7Trost, 2018 [30]AGG
G
GGGA8Hewitt, 2019 [19]GGA
G
GGGG8Li, 2019 [31]DAG
G
GIII4Kwon, 2019 [32]AAD
A
AGGA7Ahmadi and Brooks, 2020 [33]AGNA
NA
IAGA5Ahmadi and Pavey, 2020 [34]AGG
G
AAGA8Airaksinen, 2020 [35]AGG
G
GGAI 7Jun, 2020 [36]DID
G
IIIA2Franchak, 2021 [37]GGG
G
GGGA8Airaksinen, 2022 [38]AGG
A
GGAA8Madej, 2022 [20]DGI
I
ADII2Number of studies that scored G or A (out of 20) 14 19 15
 16
 17 12 10 12
General design items: 1. Provide a clear research aim, including: (1) machine learning approach, sensor, population, and specific postures and movements classified. 2. Provide a clear description of the postures and movements to be measured. 3. Provide a clear description of the development approach for machine learning or algorithm model, including a description of the target population for which the machine learning was developed. 4. This criterion was not used in this review as it was related to the conceptual framework used to define the construct measured, which was not necessary for posture and movement measured (greyed out). 5. Provide a clear description of the structure of the final machine learning or algorithm model. 6. This criterion was not used in this review as describing existing evidence on the quality of measures was not necessary for posture and movement measurement (greyed out). 7. Provide a clear description of the intended context of use for the machine learning. 8. Provide a clear description of the inclusion and exclusion criteria for the sample (e.g., clinical condition or typically developing) and characteristics (e.g., age, sex, country). 9. Provide a clear description of the method used to recruit and select sample. 10. Describe whether the sample is representative of the target population for use of the machine learning. Assessed: as good (G), adequate (A), doubtful (D), inadequate (I).


Table 2 provides the ratings of the COSMIN guideline items on criterion validity quality assessment. For the five components of criterion validity, the total number of ‘good’ or ‘average’ ratings ranged from 1/5 to 5/5. A total of ten papers scored ≥3/5; the remaining papers scored ≤2/5. Papers most commonly fell short in adequately describing the gold standard used and in providing information about missing data.


sensors-23-09661-t002_Table 2Table 2Quality assessment based on criterion validity.Author, DateCriterion ItemsNumber of Good or Adequate (out of 5)
123456
Parkka, 2010 [21]III
GI1Boughorbel, 2011 [22]DII
GI1Trost, 2012 [23]IGI
II1Suzuki, 2012 [24]AIA
II2Nam, 2013 [25]DIA
AI2Zhao, 2013 [26]AGG
GI4Goto, 2013 [27]AAA
II3Hagenbuchner, 2015 [28]IAI
GI2Hegde, 2018 [29]IAI
GI2Trost, 2018 [30]IAI
GI2Hewitt, 2019 [19]GGG
GG5Li, 2019 [31]IAA
GA4Kwon, 2019 [32]GGG
IG4Ahmadi and Brookes, 2020 [33]GGG
GI4Ahmadi and Pavey, 2020 [34]GGG
GI4Airaksinen, 2020 [35]GGG
GI4Jun, 2020 [36]AAG
GI4Franchak, 2021 [37]GAG
GG5Airaksinen, 2022 [38]GGG
GG5Madej, 2022 [20]IID
II0Number of studies that scored G or A (out of 20)111513
155
Criterion validity items: 1. Describe whether the proposed criterion can be considered a reasonable ‘gold standard’. 2. Perform the analysis with an appropriate number of participants. 3. Use contemporaneous data collection for machine learning data and ‘gold standard’. 4. This criterion (on just continuous scores) was combined with 5 (on just dichotomous scores) for this review as either continuous or dichotomous scores could be used. 5. For continuous scores: calculate correlations or area under the receiver operating curve. For dichotomous scores: determine sensitivity and specificity. 6. Report why data missing or not used. Assessed as: good (G), adequate (A), doubtful (D), inadequate (I).


### 3.2. Study Design

Table 3 details characteristics of the study design for each paper reviewed.


sensors-23-09661-t003_Table 3Table 3Details of study design as reported in each paper.Author, DateParticipants (n, Age, Sex, Clinical or Typically Developing Population, How Recruited, Country where Study Completed)Study AimSensor Information (Number of Sensors (Company Name), Hardware Type, Hardware Specifications, Sampling Rate, Sensor Location)Specific Postures and Movements IdentifiedData Collection Procedure and EnvironmentParkka, 2010 [21]n = 7, 4–37 years old (one 4-year-old child), sex not reported, clinical/TD not reported, how recruited not reported, Finland.To evaluate an activity recognition algorithm based on a decision tree classifier to automatically recognise physical activities on a portable device online, and a personalization algorithm to assist in monitoring an individual’s physical activity habits.4 sensors (Nokia). Hardware included: 3D accelerometer, sampled at 50 Hz, mounted on bilateral ankles and wrists. Six activities: lying, sitting, standing, walking, bicycling, running.Standardised tasks; volunteers performed 5 min of each activity, no detail of data collection environment.Boughorbel, 2011 [22]n = 1, 2 years old, sex not reported, clinical/TD not reported, how recruited not reported, Netherlands.To apply automatic recognition of child activities with two targeted applications: real-time automatic recognition of acute child safety (e.g., fall detection and stair climbing) and long-term activity recognition and logging to track child development and prevent child obesity.1 sensor. Hardware included: tri-axial accelerometer, tri-axial gyroscope, air pressure, sampled at 50 Hz, placed in back trouser pocket.Seven activities: walking, lying down, running, climbing stairs, falling, other.Free play; 30 min total, indoors “normal activity” suggesting home environment.Trost, 2012 [23]n = 100, 5–15 years old, evenly distributed across age range and approximately equal male and female (no specific details reported), clinical/TD not reported, how recruited not reported, Australia.To develop and test neural networks to predict children’s activity type and physical activity energy expenditure.1 sensor (Actigraph GT1M), Hardware included: an accelerometer, magnitude range 0.05–2.0 g. sampled at 30 Hz. Indirect calorimetry using Oxycon Mobile. Mounted on waist at mid-axilla line at the level of the iliac crest.12 activities classified into five distinct physical activity types: sedentary (lying down, handwriting, computer game); walking (comfortable overground walk, brisk overground walk, brisk treadmill walk); running (overground run/jog); light-intensity household activities or games (floor sweep, laundry task, throw and catch); moderate-to-vigorous-intensity games or sport (aerobic dance, basketball).Standardised tasks; Collected the 12 activity trials over two laboratory visits scheduled in a 2-week period. Each activity trial 5 min, except lying down, which was 10 min. Utilised 2 min of data from middle of trial for each activity for model developmentSuzuki, 2012 [24]n = 6, 3–5 years old, all female, clinical/TD not reported, how recruited not reported, Japan.To evaluate the accuracy of one arm accelerometer for activity recognition, the difference in accuracy between child and adult, and whether SOM has advantages over other classifiers.1 sensor (Angel band). Hardware included: accelerometer, EMG, temperature, RFID, microphone. The accelerometer was a 3-axis Wireless Tech sensor magnitude range ±17 g, sampled at 100 Hz.Mounted on upper arm.Seven activities: standing, walking, running, sitting, sleeping (lying), climb up, and climb down.Standardised tasks, each activity performed for at least 15 s. ~4 min for each participant. Unclear environment.Nam, 2013 [25]n = 3, 16–20 months old, all male, TD, how recruited not reported, Korea.To describe and evaluate an activity recognition system using a single 3-axis accelerometer and abarometric sensor worn on the waist of the body.1 sensor (SkyeModule M1-mini). Hardware included: a 3-axis accelerometer, one air pressure sensor and one near-field sensor RFID. Accelerometer magnitude range, ±2 g, sampled at 95 Hz. Mounted on the hip.11 activities: wiggling, rolling, standing still, standing up, sitting down, walking, toddling, crawling, climbing up, climbing down, stopping.Standardised tasks. Participant performed 1–2 s trial for each of 11 activities. Single home living room and kitchen environment. Zhao, 2013 [26]n = 69, 3–5 years old, ‘balanced age and gender’, TD with 20% classified as overweight/obese, recruitment reported, USA.To develop and compare multinomial logistic regression and SVM classification of physical activities among preschool children using triaxial accelerometry data.1 sensor (ActiGraph GT3x+) Hardware type: accelerometer, magnitude range ±6 g, sampled at 30 Hz. Mounted on right hip.12 activities: Sleep, watch TV, seated colouring at desk, seated video games, seated floor puzzles, play toy kitchen/blocks, ball toss and quick walking, standing active video game, dance following video instructor, aerobics following video instructor, running in place on game mat.Reclassified into six activities: sleep, rest reclining, quiet sitting play, low active play standing, moderately active play standing, very active play standing.Standardised tasks. Children wore the sensor one full day (9 a.m.–4 p.m.) and performed a series of activities in a set order, each for 10 min to 2 h duration with some free-time light activities in between.Goto, 2013 [27]n = 10, 3–5 years old, sex not stated, clinical/TD not reported, recruited via childcare centre, Japan.To develop and evaluate a single arm sensor and SOM system to classify infant activities.1 sensor (Wireless-T), Hardware: 3-axis accelerometer magnitude range 17 g, sampled at 100 Hz. Mounted on upper arm.Seven activities subcategorised into two classes: dynamic activities (walking, running, playing) and static activities (sleeping, eating, hand motion, sitting).Duration not stated. Some scenarios required of child e.g., sitting reading book and playing a puzzle.Hagenbuchner, 2015 [28]n = 11, 3–6 years old, 45% male, clinical/TD not reported, reports 9.1% were overweight, word-of-mouth recruitment, Australia.To evaluate conventional feed-forward artificial neural network with more advanced deep learning-inspired neural network for predicting physical activity types in preschool children.1 sensor (Actigraph GT3x+), Hardware type: accelerometer, magnitude range ±6 g, sampled at 100 Hz. Mounted on hip.Five classes: sedentary, light activities/games, moderate-to-vigorous activities, walking, running.Standardised tasks. 12 structured activity trials (e.g., watching TV, doing collage, playing active game) for 4–5 min each over two lab sessions within a three-week period. First visit: watching television, sitting on the floor reading, standing making a collage on a wall, walking, playing an active game, and completing an obstacle course. Second visit: sitting on a chair, playing a computer tablet game, sitting on floor playing quietly with toys, treasure hunt, cleaning up toys, bicycle riding, and running.Hegde, 2018 [29]n = 21, 11 typically developing children (mean age = 6.6 ± 1.5 years), 55% male, 10 children with cerebral palsy (mean age 6.2 ± 1.5 years), 60% male, recruitment unclear, USA.To develop a wearable sensor system for combined activity and gait monitoring in children with cerebral palsy.6 sensors, Hardware types: 1 3-D accelerometer and 5 Force Sensitive Resistor (FSR) sensors (intelink), sampled at 400 Hz. FSR sensors in insole. Accelerometer mounted on back of heel of shoe within a plastic enclosure.Four classes (each with different conditions): sitting (on child chair, on adult chair, on parent’s lap, on floor playing with toys); standing (standing still, standing while playing with toys, standing while being dressed); walk (slow walk, fast walk, run, each also completed on GAITRITE).Standardised tasks in a laboratory. Each condition completed for 2 min. When child walked on GAITRite, it was for the span of the GAITRite mat.Trost, 2018 [30]n = 11, 3–6 years old, 45% male, clinical/TD not stated—however, states that 9.1% were overweight, word-of-mouth recruitment, Australia.To develop, test, and compare human activity recognition algorithms trained on raw accelerometer signal from wrist, hip and the combination of wrist and hip in preschool-aged children. Evaluated conventional physical activity cut-point methods to activity class recognition models.2 sensors (Actigraph GT3x+), Hardware type: accelerometers, magnitude range ±6 g, sampled at 100 Hz. Mounted on hip and non-dominant wrist.Five classes: sedentary, light activities/games, moderate-to-vigorous activities, walking, and running.12 structured activity trials, identical Hagenbuchner, 2015.Hewitt, 2019 [19]n = 32, 4–25 weeks, 59% male, recruited from early childhood nurse and advertisements around university. Unclear if typically developing. Referred to as “sample of convenience”, Australia.To test the practicality of using accelerometer-based devices on an infant’s body to objectively measure tummy time and test the accuracy of manufacturers algorithm or cut points for predicting posture.4 wearable sensors (Actigraph, GENEActive, MonBaby); Hardware type: accelerometers, 2 sampled at 30 Hz, 1 at 6.25 Hz. Mounted on right hip and ankle, and chest.Three classes consisting of 12 positions: Prone floor positions (prone-on-floor attempt 1 and 2); non-prone positions (supine, left-side lie, right-side lie, cradle hold, reclined in car seat, upright against parents shoulder while parent is standing, supported sitting on lap of parent, reclined in pram); prone supported positions (being held while infant is on tummy (carer sitting or standing)); prone but lying on parent’s chest who was reclined on bean bag.Standardised tasks. 1 h testing, testing session video recorded. Infant placed in each position by parent for 3 min.Li, 2019 [31]n = 16, age 5–15 years old, sex not stated, clinical/TD likely asthmatic, unclear recruitment, however, reference dataset, BREATHE cohort, USA.Final data n = 14 (as two had substantial missing data).To develop a sensor-based integrated health monitoring system for studying paediatric asthma–specifically monitoring physical activity. To compare greedy Gaussian segmentation (GGS) with a standard fixed-size window/sliding-window approach using data from 2 HAR studies (one adult, one child) of different durations and sensor locations (just one for children).1 sensor (Motorola Moto 360); Hardware type; 3-axis accelerometer and gyroscope, sampled at 10 Hz. Mounted on wrist.Five activities: standing; sitting; lying; walking; stairs; runningStandardised tasks. 10 min for each activity, except 5 min for running. Location not stated. Randomly divided participants’ activity sessions (10 min long) into 10 subsessions. Randomly shuffled all subsessions. Kwon, 2019 [32]n = 24, 13–35 months old (‘50% one year olds’), 50% girls, recruited among visitors to a commercial indoor playroom, clinical/TD not stated, children had to be able to independently walk, USA.Final data n = 21.To describe raw accelerometer and activity count for nine activities; to evaluate the use of ML to separate ‘being carried’ from ambulatory behaviours, and to evaluate the use of ML to separate ‘being carried’ from crawling.2 sensors (Actigraph GTxX-BT) Hardware type: accelerometer, magnitude range ±6 g, sampled at 30 Hz. Mounted on hip and non-dominant wrist (left when non-dominant was unknown).Nine classes: run, walk, crawl, climb, ride-on-toy, stand, sit, stroller/wagon, and carried.Standardised tasks performed in a commercial playroom from where participants were recruited—familiar environment. Caregiver encouraged child to do the nine behaviours e.g., kitchen play for standing, block play for sitting. Mean 15 min (range 8–25 min) data per child. Mean accelerometer data per behaviour/child was 6–14 s. First and last s of activity not used. Average of 15 min of data per participant, range of 8–25 min.Ahmadi and Brookes, 2020 [33]n = 31, 3–5 years old, mean age 4.0 ± 0.9 years, 22 male, clinical/TD not stated, mainly word of mouth/local recruitment, Australia.To evaluate the classification accuracy in free-living conditions of an existing laboratory-developed ML system for preschoolers.1 sensor (Actigraph GT3x) Hardware type: accelerometer, magnitude range ±6 g, sampled at 100 Hz. Mounted on hip and non-dominant wrist.Five classes: sedentary, light activities and games, moderate-to-vigorous activities and games, walk, run.Free play. 20 min free play in home or park chosen by parent, some age-appropriate toys provided, no prompting for activities performed.Ahmadi and Pavey, 2020 [34]Identical to Ahmadi and Brookes, 2020.To evaluate ML developed on free-living data, using 1–15 s windows (1, 5, 10, 15 s), lagged and lead frames, and based on multiple sensors.Identical to Ahmadi and Brookes, 2020.Identical to Ahmadi and Brookes, 2020.Identical to Ahmadi and Brookes, 2020.Airaksinen, 2020 [35]n = 24, 7 months old (range 4.5–7.7 months, mean age 6.7 months ±0.84; 9 male), TD, recruited via larger ongoing research project, Finland.Final data n = 22.To develop a wearable sensor suit-based system to assess infant movements as early indicator of neurocognitive disorders.4 sensors; Hardware: triaxial accelerometer and gyroscope (Movesense IMU), Sampled at 52 Hz, Mounted on upper arms and legs.Iteratively developed five posture categories: prone, supine, side left, side right, crawl position. Eight movement categories: macro still, turn left, turn right, pivot left, pivot right, crawl proto, crawl commando, crawl 4 limbs (crawl 4 limbs omitted as only one recording utilised category).Standardised tasks. In clinic-like settings for 30–60 min.Physiotherapist encouraged a range of postures and movements by play without touching infant. Movements collected while infant was placed on a foam mattress. Mean 29 min of data collection (range 9–40 min). Total of 12.1 h recorded.Jun, 2020 [36]n = 10, 2–720 days old, 7 male, potentially some clinical population as authors refer to “newborn’s physical condition and other medical devices attached”, recruitment strategy unclear, Korea.To develop a method which can classify activity types from sensor signals, whether subjects are asleep, how strong movements show, and whether external forces affect them.1 sensor; Hardware type; triaxial accelerometer and thermometer, sampled at 40 Hz. Mounted to clothing on “upper chest area”—any area of the chest and above clothes without giving precise position and clothes condition.Three levels of classification: sleeping/non-sleeping; sleeping/active movement/external force movement; and sleeping/strong movement (struggling or crying in agony)/weak movement (awake and moving in comfortable state)/external force movement.Unlcear activities. No detail on what specific activities the newborn performed. Video lengths ranged from 5–150 min in duration (700 min total). Some participants did not include both sleep and awake states.Franchak, 2021 [37]Laboratory study: N = 15, 6–18 months old, eight female, TD unclear, recruited via social media advertisements and local community recruitment events, USA.Home data collection case study: N = 2, 10.5–11 months old, sex unclear. Likely from the lab study; however, unclear in reporting. Note neither infant could walk independently; however, both could stand, cruise along furniture, and walk while supported with a push toy or caregiver assistance.To develop and validate a classification system using infant-worn inertial sensors to classify typical postures and movements in an infant’s day, to assist with monitoring infant movement behaviours in the home environment. Aimed to assess whether the method could accurately detect individual differences in how much time infants spend in different postures, to characterise everyday movement experiences and their potential for developmental impact.Laboratory study: 3 sensors (MetamotionR IMU); accelerometer and gyroscope, sampled at 50 Hz. Mounted on right hip, thigh, and ankle.Home data collection case study: four Biostamp IMUs (accelerometer and gyrosocope) Sampled at 62.5 Hz, embedded in pair of customized infant leggings–placed bilateral hip and ankle.Five body positions: supine (lying on back), prone (lying flat on stomach or in crawling position), sitting (sitting on a surface with or without support from caregiver, the highchair, or on caregiver’s lap), upright (standing, walking, or cruising along furniture), held by caregiver (carried in caregiver’s arms, excluding times they were seated on caregiver’s lap).Standardised tasks in a laboratory: 10 activities (assisted or unassisted)—standing upright, walking, crawling, sitting on the floor, lying supine, lying prone, held by a stationary caregiver, held by a caregiver walking in place, sitting restrained in a highchair. Completed each activity for 1 min, total session lasting 10 min, followed by free play to allow for spontaneous body positions.Standardised tasks in the home environment:Experimenter guided caregiver via phone through a set of procedures to elicit different body positions—tasks the same as the laboratory tasks, completed each activity 1 min, followed by 10 min of free play. Following free play, infant and caregiver went about day as normal wearing IMUs for approx. 8 h—video recording was for up to 180 min during this time.Airaksinen, 2022 [38]n = 59, 4.5–19.5 months old, sex not stated, n = 38 neurodevelopment low-risk term born, n = 10 mild prenatal asphyxia, n = 11 prematurity, 4 later found to have neurodevelopmental condition (left out of some modelling), recruited from hospital, Finland. To develop a wearable sensor suit-based system to assess infant movements, across infant milestones from lying to walking as early indicator of neurocognitive disordersIdentical to Airaksinen 2020.Iterative developedposture/movement matrix of five different postures with four movement conditions for each posture:postures (supine, prone, crawl, sitting, standing) and movements (still, proto, elementary, fluent, other, carrying).Free play at home (n = 40) or home like clinic (n = 24). Average data 67 min (range 18–199 min) total recording time 71 h and 30 min. Children encouraged to free play with little adult interference, differences in environment/play opportunities. Participants collected at home instructed to play for at least 1 h.Madej, 2022 [20]n = 10, 4–40 years old (mean 24 years ± 14 years), 7 men, unclear if clinical/TD, unclear recruitment, Poland.To determine whether there is a difference in physical activity assessment between wrist-worn sensor on the dominant and non-dominant arm and between lower back and hip-worn sensor.2 sensors (Mbient Lab Meta-motion IMU’s). Hardware type: accelerometer, gyroscope and magnetometer. Accelerometer range was ±16 g at 100 Hz, magnetometer range as ±1300 uT at 25 Hz and gyroscope was ±2000 st/s. The 4 sensors were mounted on both wrists, lower back, and hip on dominant hand side (upper limb collected separately to low back and hip).Nine activities: jumping, rotating, running, walking, walking on tiptoe, clapping hands, standing still, sitting still, and dancing.Standardised tasks. All activities performed for 15 s with 5 s standing between and done twice—once with two wrist sensors and once with two lower-body sensors. 10 s of each activity was used for analysis.EMG = Electromyography. HAR = human activity recognition. IMU = inertial measurement units. ML = machine learning. RFID = radio frequency identification. SOM = Self-Organising Map. SVM = Support vector machine. TD = typically developed.


#### 3.2.1. Participants

Reviewed papers acquired data for model development and evaluation from 1 [22] to 100 [23] participants, with most papers (n = 14) having less than 25 participants. Most of the papers included an even number of male and female participants. The age range varied widely: eight papers included children under the age of 3 years, four papers included children aged between 3–5 years [26,27,33,34], and the remaining 12 papers included children over the age of 5 years or adults with a small set of children who fit the inclusion criteria. Some of these only included one participant fitting the inclusion criteria or were unclear in how many participants of each age were within the sample. Most papers did not explicitly state whether children were typically developing; however, 17 papers appeared to develop their models on data acquired from typically developing children without clinical diagnoses. Of the three papers that included children with clinical diagnoses, one included children with asthma [31], one included both typically developing children and children with cerebral palsy [29], and one appeared to include children with unknown clinical conditions, where the paper refers to “newborn’s physical condition and other medical devices attached” [36].

#### 3.2.2. Aims

Eleven of the reviewed papers aimed to develop systems that would allow for physical activity or general activity monitoring, to be used in understanding child development and preventing lifestyle diseases such as obesity. However, the aims of the remaining nine papers varied widely. Specifically, of the nine remaining papers, two evaluated systems in a different environmental context (e.g., free-living) [33,34]. Two papers had the specific aim of evaluating infants’ movements as early indicators of neurocognitive disorders, and one focused on evaluating methods for measuring the time infants spent in prone lying postures (i.e., tummy time) [35,38]. One paper was focused on the prevention of falls [22], and two papers focused on the development of a system to measure postures and movements in children [27,36].

#### 3.2.3. Sensor Information (Type, Sampling Rate, Number of Sensors, Locations)

A range of commercially available and custom-built wearable movement sensors were utilised. Eleven of the papers developed/evaluated models using only accelerometer data, whilst five papers utilised inertial measurement unit sensors [20,31,35,37,38].Four papers combined accelerometer data with other types of non-sensor-based and sensor-based data, including calorimetry (n = 1 [23]), air pressure sensors (n = 2 [22,25], and force pressure sensors (n = 1 [29]). One paper compared approaches using different numbers of accelerometers and additional data [19]. Eleven of the papers utilised a single accelerometer, and accelerometers were located in a range of locations: upper chest (n = 1 [36]), upper arm (n = 2 [24]) wrist (n = 1 [31]), waist (n = 2 [23,25]), hip (n = 2 [26,28]), back pocket of trousers (n = 1 [22]), and shoe (n = 1 [29]). Four of the papers used two accelerometers, and these were located on the hip and wrist [30,32,33,34]. One paper used three accelerometers located on the hip, thigh, and ankle [37]. This paper also described a second home-data collection phase of their model development and evaluation, where they used four lower limb accelerometers (bilateral hip and ankle) based on the results of the laboratory-based study [37]. The remaining four studies used four accelerometers with a range of location combinations, which all combined upper-body- and lower-body-mounted accelerometers [19,21,35,38].

#### 3.2.4. Postures and Movements Measured

The postures and movements most commonly included were lying, sitting, standing, walking, and running. For papers that included lying, half specifically focused on the orientation that the child was lying in (e.g., prone/supine/side lying), and the other half did not differentiate the orientation. All papers that differentiated lying included children under the age of three years old. For papers that included sitting, the sitting data used for model development sometimes included sitting on varied surfaces and in different conditions; however, many did not report the specific posture the child was sitting in, or if the child self-selected their sitting posture when the data was collected. Additionally, very few of the models developed included sitting and standing with and without movement, and those that did focused on physical activity intensity classifications. Five of the reviewed papers developed machine learning models that classified both specific movements and physical activity intensity, where the specific postures and movements included were walking and running. More diverse and child-specific movements such as crawling (n = 5 [25,32,35,37,38]), climbing (n = 1 [22]), and jumping (n = 1 [20]) were less commonly identified.

#### 3.2.5. Data Collection Methods

The majority of papers explicitly stated data were acquired in a laboratory-based environment (n = 7), whilst others (n = 6) did not state the location, but the methods suggested that it was within a laboratory. One paper [37], conducted a laboratory study which was repeated (with some modifications to sensor locations) in a home environment. Within the home environment, both prescribed activities and free-living activities were collected. The remaining papers (n = 7) collected data within various ”free living” environments, which included indoor play centres, healthcare clinics, childcare centres, the child’s own home, and a park. Tasks were performed during data collection for between 15 s and 5 min each.

### 3.3. Classification Model Development

Table 4 details the classification model development for all included papers, with a focus on window size, feature extraction methodology, machine learning approach applied, model development, and validation approaches.

Data sampling windows used were either overlapping (n = 7), non-overlapping (n = 8), or the use of overlap or not was not specified (n = 5, [19,20,21,22,36]). Four of the studies compared models using different window sizes ranging from 0.2 s to 60 s [23,28,31,34]. Window size was not reported in four papers [20,22,24,27].

A range of different features were extracted for model development, and these were typically amplitude and frequency domain features. This was performed for all non-deep learning models (n = 15), as well as the first model of the first Airaksinen paper [35]. For the second model of the first paper [38] and the second paper [35], the time signal was input into supervised deep learning models as well as into an unsupervised deep learning model (n = 1).


sensors-23-09661-t004_Table 4Table 4Details of classification model development as reported in each paper.Author,DateApproachWindow Details (Size, Overlapping or Non-Overlapping)Feature ExtractionMachine Learning Method UsedParkka,2010 [21]Non-MLNon-overlapping 5 s windowsFour features obtained: intensity of highest peak in power spectral density; average signal; signal spectral entropy; and signal variance. Only ankle accelerometer vertical axis data used for computing feature signals.N/ABoughorbel, 2011 [22]MLWindows size not specified.Four first-order (vector magnitude of accel, vector mag of gyro, normalised Z of accel, measured air pressure) and for each of these, five second-order features calculated (moving average, moving variance, moving RMS 0.1–2 Hz, moving RMS 2–4 Hz moving average slope) i.e., 20 s-orderLinear (LDA), Quadratic (QDA), and AdaBoost classifiers.Trost,2012 [23]MLSequence of non-overlapping windows of 10,15, 20, 30, and 60 s durationSix features obtained: 10th, 25th, 50th, 75th, and 90th percentiles of second-by-second counts and the lag one autocorrelation.Feed-forward neural network with a single hidden layer.Suzuki,2012 [24]MLFeatures extracted by moving windows with 50% overlap (window size not specified).Five features: mean, standard deviation, energy, correlation, frequency domain entropy.J48, Naïve Bayes, NBTree, Random Forest, RandomTree, REPTree, and self-organizing map (SOM).Nam,2013 [25]MLWindow size of 256 with overlapping at 128 at 95 Hz.Features obtained for x, y, z, and derived horizontal and vertical traces. Five time domain and two frequency domain features extracted for each.Naïve Bayes, Bayes Net, Support Vector Machine, k-Nearest Neighbour, J48 Decision Tree, Decision Table, Multilayer Perceptron, Logistic Regression.Zhao,2013 [26]MLNon-overlapping windows of 60 s.Counts in x, y, and z, vector magnitude, position, steps, lag/lead values.k-means used to identify clusters of activities—used as evidence to recategorize the data; MLR and SVM used to classify; 58 classifier models built. Models including and not including sleep.Goto,2013 [27]MLMoving window 50% overlap (although window size not specified).Two-stage process. Phase 1 determined if static or dynamic, using standard deviation and energy of each 3-axis. In phase 2, classified for static (sleeping, eating, hand motion, sitting) and dynamic (walking, running, playing) by adding movement of gradient to features.Self-Organising Map (SOM) used for both phases.Hagenbuchner,2015 [28]MLNon-overlapping window sizes of 10, 15, 20, 30 and 60 s.Same features as Trost 2012 study.Multi-layer Perceptron Network (MLP), Self-Organising Map (SOM), and third employed SOM as first layer followed by MLP.Hegde, 2018 [29]MLNon-overlapping window size of 2 s.12 features extracted: six from each sensor: mean of sum of all five pressure sensor data from shoe (P_Sum), standard deviation of P_Sum, mean of resultant acceleration, standard deviation of resultant acceleration, number of mean crossings of P_Sum, number of mean crossings of resultant acceleration. Multinomial Logistic Discrimination.Trost,2018 [30]MLNon-overlapping 15 s windows.18 time and frequency features extracted: mean, SD, minimum, maximum, interquartile range, percentiles (10th, 25th, 50th, 75th, 95th), coefficient of variation, signal sum, signal power, peak-to-peak amplitude, median crossings, dominant frequency between 0.25 and 5.0 Hz, magnitude of dominant frequency between 0.25 and 5.0 Hz, and signal entropy between 0.25 and 5.0 Hz.Random Forest and Support Vector Machine each used for hip, wrist, and hip and wrist (total of six classifiers).Hewitt,2019 [19]Non-MLWindow size of 1 s for Actigraph and GENEactiv and of 1/(5–7) s for MonBaby.Actigraph: custom built Excel macro designed by Actigraph. Used specified X and Y-axis cut points hip Actigraph: X-axis > 0.7 g and Y-axis > −0.1 g for prone on floor; x axis > 0.7 g and Y-axis < −0.1 g for prone supported. For Actigraph ankle X and Z-axis cut points used; X-axis > 0.35 g and Z-axis > −0.45 g for prone.GENEActiv: algorithm developed by Activinsights Ltd.—formed by classifying each position for a scatter plot with rotation on the X-axis and elevation on the Y-axisMonBaby: 360-degree angle determined from X, Y, and Z-axes, 360-degree angles less than 134 degrees classified as non-prone. Prone on floor and prone supported positions were determined using Z-axis cut point of <−0.10 g.N/ALi,2019 [31]MLGreedy Gaussian segmentation (GGS) compared to fixed-size non-overlapping windows of 0.2 s, 0.8 s, 3 s, 8 s, for HARuS data set, and 12 s and 40 s added for BREATHE.168 features: Six statistics (arithmetic mean, SD, median absolute deviation, minimum, maximum, and entropy) on 14 signals and on both the time and frequency domains (6 × 14 × 2 = 168).XGBoost. SVM and Random Forests were also trained but only presented in supplementary material.Kwon,2019 [32]MLSegmented accelerometry into non-overlapping windows of 5 s. Only windows with single behaviour included.Activity counts from accelerometer vertical, horizontal, perpendicular axis and vector magnitude, 30 time-domain (e.g., mean, SD, skewness) and 48 frequency-domain features (from FFT) were extracted.Used Random Forest but only to differentiate between “carried” and “ambulation”.Ahmadi and Brookes, 2020 [33]MLNon-overlapping sliding window of 15 s and considered ML including and not including mixed windows.Same 18 features as Trost, 2018.Used earlier Trost Random Forest and SVM models to test their efficacy on free-living behaviours.Ahmadi and Pavey,2020 [34]MLNon-overlapping windows of 1, 5, 10, 15 s. Mixed/not mixed activities.Two sets of features were extracted: Base features (same 18 time- and frequency-domain features as prior work) and base plus temporal features (considering preceding and following windows, i.e., lead and lag, resulting in 5 additional features).Random Forest.Airaksinen, 2020 [35]MLData windowed into 120-sample frames (2.3 s at 52 Hz) with 50% overlap between subsequent windows.Two approaches compared, only first used features. 336 features extracted, with 14 features (variance, max amplitude, min amplitude, signal magnitude area, energy, interquartile range, skewness, kurtosis, largest frequency component, weighted average frequency, frequency skewness, and frequency kurtosis) for each of 24 data channels.First was a Support Vector Machine. Second was a convolutional neural network with 3 stages—(1) sensor module to extract low-level features, (2) sensor fusion model for combining sensor features into common high-level features, (3) time series model for temporal modelling of high-level features. Jun,2020 [36]ML160 elements, each length 3. Uses sliding windows of 40 elements, which corresponds to 1 s long. Two consecutive have 75% in common. Some data preprocessing but note, is essentially feeding sensor data into model.N/AUnsupervised deep learning. Model divided into autoencoder and k-means clustering algorithms. The first finds a minimal space that can reproduce signal. This minimal space is then clustered.Franchak, 2021 [37]MLOverlapping moving windows of 4 s extracted every 1 s. Excluded windows where <75% single position.204 features extracted. Ten summary statistics for each combination of 3 sensor locations, 2 sensor signals, and 3 axes, resulting in 180 features: mean, standard deviation, skew, kurtosis, minimum, maximum, 25th percentile, 75th percentile, and sum. Sum and magnitude of movement across axes within each sensor. Correlations and difference scores between each pair of axes within a sensor and between each pair of sensors for a given axis.Random Forest.Airaksinen, 2022 [38]MLSame overlapping window as previous work, i.e., data windowed into 120-sample frames (2.3 s at 52 Hz) with 50% overlap between subsequent windows.N/ASimilar CNN model as one used in Airaksinen 2020.Madej,2022 [20]Non-MLFeatures calculated for each accelerometer measurement.35 features extracted (mean, SD, skewness, kurtosis, energy, activity, mobility, complexity, and spectral purity index for each axis, 3 correlations between axes and the mean over 3 axes of mean, SD, skewness, kurtosis, energy)N/AML = machine learning. Non-ML = Not machine learning.


### 3.4. Model Accuracy

Table 5 and Table 6 detail the accuracy of the developed models.

Human coding by direct observation or later video observation was most commonly used as the gold standard comparison; however, several papers did not clearly report on the method for the collection and annotation of the comparison data.

A range of approaches were used in developing and validating model accuracy. These included leave-one-subject-out cross-validation (n = 6), 10-fold cross-validation (n = 3), and 3-fold cross-validation (n = 1 [22]). One paper [31] split the data set into a training set and test set, where the model was trained on 12 participants and tested on the remaining two. One paper split the data into three evenly sized data sets, one for training, one for validation, and one for testing [23]. The remaining machine learning model papers used a combination of validation approaches.

The majority of papers used confusion matrices to determine accuracy, although there was little consistency of what was included in the confusion matrices. For example, recall and precision (%) [31], prevalence, sensitivity, and positive predictive value [37], or just ‘accuracy’ [25]. How each of the statistics reported were calculated was often not explicitly stated.

Table 6 summarises the accuracy reported for the models validated in each paper. There was a wide range reported for overall accuracy of the models (i.e., the degree of accuracy considering all postures and movements included in the model) of 59–97%. Further, a large accuracy range commonly existed in models to detect each specific posture and movement (see Table 6). For example, models were able to detect sitting with a range of 53–100%, walking with a range of 9–99%, and running with a range of 18–100% accuracy. Five of the papers only reported overall accuracy or did not report posture- or movement-specific accuracy [19,20,24,31,32].


sensors-23-09661-t005_Table 5Table 5Details of classification model accuracy.Author, DateGold Standard Used for Comparison in Development (Including Inter-Coder Reliability Testing)Validation Approach UsedAssessment of AccuracyAccuracy of the SystemParkka,2010 [21]No detail provided.No separate testing data set. Leave-one-subject-out (LOSO) cross-validation.Confusion matrix.Without personalisation: overall accuracy 86.6%, walking only accurately recognised 48% of the time.With personalisation: overall accuracy 95%. Model performance consistently poorer on single 4-years-old participant (74% accuracy).Boughorbel, 2011 [22]Synchronised video ‘manually annotated’.Data from single child randomly split into training and test data sets (size not specified). 3-fold validation was employed.Not stated.Using first-order features, mean accuracy was 38 ± 1.5%.Using second order 97.8 ± 0.2%, using only accelerometer data 79.9 ± 1.5% (full confusion matrix provided).Trost,2012 [23]No detail provided.Data randomly divided into training, validation, and test sets of approximately the same size. 10 such random splits were performed.Confusion matrix (proportion of time segments correct identified).Walking 92–94% accuracy (accuracy increased with increase in window size). Running 74–79% accuracy (accuracy increased with increasing window size). Running trials most commonly misclassified as walking. If running and walking combined into “locomotion”, 96% accuracy.Suzuki,2012 [24]Voice recording during data collection.No separate testing/training data sets. Three validations undertaken against child and adult group. Self-validation trained by either child or adult data.K-fold cross-validation trained by same group data. Test data validation using other group data.Only overall accuracies reported, not posture/movement specific. No confusion matrices presented.Self-validation: Child 88 (REPT)-100% (SOM, RF, RT)Adult 93 (REPT)-100% (SOM, RF, RT)Cross-validation:Child 36 (REPT)-71% (RF)Adult 50 (REPT)-76% (RF)Test:Child 31 (NBT)-45% (C4.5)Adult 40 (C4.5)-50% (REPT)Nam,2013 [25]Simultaneously video recorded and later annotated.1538 samples were collected from one baby as training data, other samples collected were used as test data set.10-fold cross-validation.Precision, Recall, F measure reported.Accuracy for each movement reported.Confusion matrices for each approach reported.Recognition accuracy reported for 8 different methods; however, how accuracy is calculated not specified—likely from confusion matrix.Best performing: 95% for KNN and Decision Tree.Precision and recall: MLP kNN and Decision tree > 94%.Barometer data reduces false alarms for climbing up and down.Zhao,2013 [26]Staff minute-to-minute observation while children in room.No separate testing/training datasets; 10-fold cross-validation. Reported error rates as measure of accuracy. Confusion matrix reported.Error rate: Number of observations that had been incorrectly classified in activity divided by number of observations of given activity.Overall error rate with sleep: MLR ~30%; SVM ~26%. Without sleep: SVM 21%; ‘overall’ 16%.Similar activities with close rankings more difficult to classify than dissimilar activities.Goto,2013 [27]Video recorded of child doing activities and synchronised to accel data. Types of activities discriminated by checking the time of acceleration data and video recorded activity.No separate testing/training data sets; 10-fold cross-validation. Reported classification accuracy.Not stated.Mean 65% with range of 47% (eating) to 99% (sleeping).Hagenbuchner,2015 [28]Not stated.No separate testing/training data sets. LOSO cross-validation.Confusion matrix.60 s window: SOM 54%; MLP 70%; DLEN 83%. 30 s window: SOM 53%; MLP 64%; DLEN: 76%. 10 s window: SOM 52%; MLP 61%; DLEN 72%.Hegde,2018 [29]Smartshoe data was manually annotated, labelling the type, start, and end of each activity. No detail of what was the gold standard reference informing this.No separate testing/training data sets. LOSO cross-validation. Confusion matrix.Typically developing children average accuracy 96.2%. Children with cerebral palsy average accuracy 95.3%.Trost,2018 [30]No mention of gold standard; however, also compared to accelerometer cut-point methods (separate analysis).No separate testing/training data sets. LOSO cross-validation.Overall recognition accuracy (% of 15 s time windows correctly classified), agreement between predicted and observed class label evaluated by calculating weighted kappa coefficients. Compared to cut-point methods using sensitivity and specificityMean overall accuracy for Random Forest: Hip 80%; Wrist 78%; Combined 82%. Mean overall accuracy for SVM: Hip 81%; Wrist 80%; Combined 85%. New classifiers outperformed traditional cut-point methods for classifying PA levels.Hewitt,2019 [19]Direct observation captured on video recording of whole session. Single observer coded each second of video. One randomly selected video analyzed by four observers to test interrater reliability (97.5%).Not ML, so no testing set needed. Time spent in each position class was evaluated for each device against gold standard to determine percentage accuracy.Based on number of seconds recorded compared to direct observations.GENEActive: Prone on floor 95.4%; non-prone 98%; prone supported 52.2%. Actigraph Hip: prone on floor 90%; non-prone 99.9%; prone supported 63.6%. Actigraph Ankle: prone on floor 87.9%; non-prone 96.3%; prone supported 53.3%.MonBaby: prone on floor 79.2%; non-prone 99.9%; prone supported 66.1%.Li,2019 [31]Not stated.Multiple data sets tested but each divided into ~90% for training and ~10% holdout testing. Confusion matrix with precision and recall reported.Not stated.Overall average recall when using GGS was 73%. Overall averaged precision when using GGS was 86%. Instantaneous accuracy from XGBoost using GGS was 79.4%. Highest fixed-size window accuracy was 72.7%.Kwon,2019 [32]GoPro video recorded. Three coders independently coded first four participants using draft coding scheme; after discussion and revision, two coders independently coded rest with 96% concordance. Accel and video synched using visual inspection of active/still.No separate testing/training data sets; 10-fold cross-validation used to identify hyperparameters. LOSO cross-validation used to evaluate classifier performance.Not explicitly stated and not reported for each PAM. No confusion matrix.Carried vs. ambulation classification 89% from hip. No full confusion matrix reported. Only accelerometer descriptives for each behaviour and hip and wrist reported.Ahmadi and Brookes, 2020 [33]GoPro HERO 5 video of session, human-coded in two stages:(1) Five classes (sed, light/games, mv/games, walk, run)(2) 23 activity types (e.g., sit still, sit with upper limb movement). Some activities only performed by one participant, but classes performed by at least 28. Dual coding of five participants gave intraclass correlation coefficient for activity type of 0.912 and 0.927 for activity type.Model already developed, therefore classification was based on testing the accuracy of the model derived from lab data on field data. Overall accuracy, unweighted kappa, only done for classes. Confusion matrix for class and specific activities.Confusion matrix.Overall wrist RF 59% and SVM 59%, fair agreement (kappa = 0.37). Poor at walk SVM 12% RF 15%; however, improved to 44% and 46%, respectively, when windows with multiple classes were removed. Overall hip RF 69% and SVM 66%, moderate agreement (0.45–0.48). Also poor at walk SVM 9% and RF 11%; however, improved to 29% and 33% when windows with multiple classes were removed. Erist reduced ~20% from lab study and hip ~15%.Ahmadi and Pavey,2020 [34]Same as Ahmadi and Brooks, 2020. Cohen’s unweighted kappa statistic for activity class was 0.86, again taken from two researchers independently coding five randomly selected videos.No separate testing/training data sets. LOSO cross-validation; 3 × 2 × 4 repeated measures ANOVA used to examine effects of sensor placement, feature set (base vs. temporal features), and window size on F scores.F-scores were used to assess the accuracy of each model. Confusion matrices.F scores for best-performing wrist and hip model were 81% and 86%, respectively. Shorter windows decreased accuracy. Lag/lead did improve accuracy for models trained on wrist data on 1, 5 and 10 s windows, and combined hip and wrist for 1 s windows, but not for any other models. Multiple sensors and feature fusion did not improve accuracy. Hip better than Wrist. F scores: Wrist 62–77%; Lag/lead 69–81%; Hip 71–84%; Lag/lead 76–86%; Both 72–84%; Lag/lead 77–86%.Airaksinen, 2020 [35]Video by Gopro (n = 14) or iPhone (n = 10). Independently annotated by three researchers. Interrater agreement tested with Fleiss kappa score, yielding 0.923 for posture and 0.580 for movement. Note, developed a novel iterative annotation refinement (IAR) method to resolve ambiguities in the training data by combining human- and machine-generated labels in a probabilistic fashion during model development.No separate testing/training data sets. LOSO cross-validation.Confusion matrices.Classification accuracies of posture generally comparable between CNN and SVM. CNN better performance with several movement categories (CNN 5–10% better performing based on F score).For frames where all three humans rated the same (accuracy): posture 99% and movement 91%. For all frames (accuracy): posture 98% and movement 82%.Two-sensor data (1 arm, 1 leg) similar accuracy to four-sensor data but single sensor significantly worse (two-sensor posture 94%, movement 78%, four-sensor posture 95%, movement 79%, one-arm-only posture 71%, movement 70%, one-leg-only posture 90%, movement 68%). ML comparable to human consistency.Jun,2020 [36]Video recorded during data collection using webcam connected to data acquisition computer at 30 fps and time-synchronised with sensor data. Used for ground truth reference. Coded by two independent coders and results subsequently compared to adjust different labels after consultation. Used iterative annotation refinement to fix the often-substantial interrater disagreementsModel trained on one participant (125 min video and sensor signals). Chose not to train on random selection of data from all subjects, stating there was a high probability that activities with low frequency, such as movement by external force, would not be included in training data.Data for remaining nine subjects used to test data. Reported F1 score, precision, and recall and balanced accuracy for each activity.Recall, balanced accuracy.Overall balanced accuracy 96%, ranges from 95–97%.Franchak, 2021 [37]In laboratory study, video recorded via handheld camcorder. Each video coded in its entirety by two coders; interrater reliability determined as overall agreement 97.6% and kappa of 0.966. Home data collection case study used 360-degree camera coded by single coder using same categories as laboratory study.Individual model trained on 60% of data from individual and tested on remaining 40%. Group model LOSO cross-validation. Report classification accuracy. For home data collection case study,data from guided session combined and split into training and testing sets (60/40).Sensitivity (i.e., proportion of actual occurrences that were correctly predicted). For home-based data collection case study.Predicted positions from ML model to actual coded positions in testing data, as well as prevalence, sensitivity, and positive predictive value for each body position.Correlations between actual and predicted positions for all available data.Individual models: overall accuracy averaged 97.9% (SD 2.37%). Group model: overall accuracy averaged 93.2% (SD 0.053%). Home-based data collection case study: overall accuracy 85.2–86.6%.All available data correlations r = 0.911–0.976.Airaksinen, 2022 [38]Video 18–74 min in only n = 41 infants. Annotated by two (n = 9) or three (n = 32) independent researchers. Fleiss Kappa interrater reliability of 0.95 of postures and 0.6 of movements. Also used previously developed iterative annotation refinement (IAR) method to resolve ambiguities in the training data by combining human- and machine-generated labels in a probabilistic fashion.No separate testing/training data sets. A number of classification activities reported which used either 10-fold or LOSO cross-validation.Confusion matrices.Active carrying vs. not 97%, carrying vs. not 99%. Posture overall kappa 0.93.Movement overall kappa not given, but kappa for each posture and movement shown in a figure.Madej,2022 [20]Manually labelled offline (not explicitly stated what was used as the reference).Mean activity vector distance used to conclude whether the constructed feature vector allowed the authors to distinguish between the analysed activities. Euclidean distance was averaged over subjects for each sensor separately, then for all sensors in selected IMU and configuration.Not stated.Best result accelerometer and magnetometer on non-dominant arm (trace of minimum distances matrix = 8), worst was gyroscope on lower back and magnetometer on hip (trace = 4). Conclude no differences between wrists, nor between low back and hip.



sensors-23-09661-t006_Table 6Table 6Summary of classification model accuracy for each posture and movement, rows ordered by combined studies’ sample size and columns ordered by paper’s chronological order.
Range across Papers (n = Sum of Sample across Studies)Parkka 2010, [21]n = 2, (4–8 Years) *Boughorbel2011, [22]n = 1,(2 Years)Trost 2012, [23]n = 100(5–15 Years)Nam 2013, [25]n = 3 (16–20 Months)Zhao 2013, [26]n = 69(3–5 Years)Goto 2013, [27]n = 10 (3–5 Years)Hagenbuchner2015, [28]n = 11, (PreSchool)Hegde 2018, n = 21, [29](Healthy Children)Trost 2018, [30]n = 11 (3–6 Years)Ahmadi and Brooks [33] 2020, n = 31 (3–5 Years)Ahmadi and Pavey 2020, [34]n = 31, (3–5 Years)Airaksinen 2020, [35]n = 22 (7 Months Old)Franchak2021, [37]n = 15 (6–18 Months)Airaksinen 2022, [38]n = 59 (7 Months Old)Posture/Movement














Walk9–99% (n = 191)70–88%93%92–94%81–97%
58%36–73%97–99%61–63%9–46%65–81%


Run18–100% (n = 166)77–99%9775–79%

69%18–73%
68–75%66–100%68–88%


Intensity classification51–89% (n = 153)

58–98%


51–91%
69–93%57–84%68–93%


Lying prone67–98% (n = 96)










98%67%98%Lying Supine87–97%(n = 96)










97%97%87%Stand66–100% (n = 86)74–95%100%
66–98%


90–92%




96%Carrying45–99% (n = 84)











45%97–99%Pivot62–66% (n = 81)










63–66%
62%Crawl65–84% (n = 81)










65
84%Side lie77–81% (n = 81)










77–81%
78%Sit53–100% (n = 69)74–95%

78–96%74%53%
95–99%



88%93%Rolling65–99% (n = 62)


94–99%








64%Crawl position77–100% (n = 25)


77–100%






88%

Still85% (n = 22)










85%

Turn58% (n = 22)










58%

Crawl commando60% (n = 22)










60%

Lying–position differentiation95–100% (n = 3)74–95%95%











Climb55–96% (n = 3)


55–96%









Climb stairs99% (n = 1)
99%











Falling100% (n = 1)
100%











Note: Five papers did not report accuracy, so these were excluded from this table; accuracy = ‘accuracy’ output from confusion matrix. Colour coding < 25 summed sample in green. * Note a subset of the population was reported for one study (Parkka 2010, n = 2, age 4–8).


## 4. Discussion

The aims of this systematic review were to determine how young children’s postures and movements have been objectively classified and measured using accelerometer hardware and accompanying software, and the accuracy of current systems. The review identified 20 peer-reviewed journal papers, 17 of which reported customised machine learning-based algorithms, and three of which reported simpler, human-defined prediction approaches. While the quality of papers varied greatly, over half scored very well across study design and concurrent validity items. This review highlights the diversity of approaches that have been used to objectively capture children’s posture and movement and the impact this had on the reported accuracy. The results highlight that there is currently little consensus on: (1) which postures and movements to record, (2) the participant sample, (3) the study type/developmental approach, (4) the hardware, (5) the software, and (6) the validation approaches used. The synthesis below includes recommendations for each of these factors to help guide future development.

### 4.1. Posture and Movement

The synthesis of the 20 peer-reviewed papers in this review highlights that there is little consistency in which activities have been selected to objectively classify in young children. Indeed, there were over 30 postures and movements targeted across the included papers, with only some overlap between studies. As such, there is a need to establish a consensus on the types of postures and movements to assess, a finding echoed even in a recent scoping review that summarised how postures and movements had been objectively measured in healthy adult populations [14]. The results highlight an emphasis on measuring different lying postures in non-ambulatory-aged children (<3 years old) [19,36] which aligns with the evidence linking time spent lying in different positions with important developmental milestones in this age group. However, for older age groups (>3 years old), there was considerable diversity in movements and postures measured. The most commonly measured postures and movements across the included papers were lying, sitting, standing, walking, and running. These align well with the most frequently reported movements and postures measured in adults [14]. Few papers examined child-specific movements (e.g., crawling [25,37] and climbing [22,24,25,32]) and child-specific adaption of postures (e.g., different types of sitting such as kneeling and side sitting [29]). This might be due to most papers (n = 19) using a prescribed, standard set of tasks, that may not be reflective of free-living conditions. Even in studies examining postures and movements, physical activity intensity also remained frequently studied, which was justified through established links with childhood health [39]. Future research should continue to focus on the postures and movements that have been identified as important, while also considering diversification to more child-specific variations of these postures and movements.

### 4.2. Participant Sample

The number of participants evaluated in the reviewed papers ranged from 1 to 100 participants. However, most studies involved fewer than 25 participants, and studies rarely reported any sample size justification. This is a weakness consistent with research conducted in adults, with most samples including approximately 20 participants [14]. It is accepted that the generalisability of models developed on small cohorts is limited, despite them often reporting very high levels of accuracy. For example, the highest and most consistent accuracy reported for predicting walking was in a study with only one participant (99–100%), while the studies with the largest samples had much greater variance in the reported accuracy (e.g., 58–100%, n = 100, [23]). If the goal is to utilise the models beyond the sample they are developed on, larger sample sizes are required that are representative of the intended application population. While most of the samples were balanced for sex, it remains unclear if there is a specific effect of sex on the objective measurement of posture and movement in children from these studies. Similarly, there was a range of childhood age groups studied; however, none of the included studies specifically investigated the influence of child age on the objective classification of postures and movements. A deeper understanding of the influence of child age on prediction accuracy is warranted, as children’s postures and movements change with age (e.g., the change in gait patterns from toddlers to preschoolers). It has also been shown that a model developed on adults may not perform well on children [24]. Thus, future research should specifically investigate the influence of both sex and age on model predictions. Lastly, most of the studies involved only typically developing children (19 out of 20 papers). The success of any developed objective classification systems should be specifically checked on populations with atypical developmental profiles, for example, children that are known to move differently to typically developing children, such as children with cerebral palsy [40].

### 4.3. Study Type/Development Approach

The twenty included papers were all aimed at methodological advancement and therefore all utilised a validation study design. More than half collected the data in a laboratory-based, controlled manner (i.e., with a prescribed set of activities). This approach is common when developing methods, with most similar work on healthy adults also collected in a laboratory environment using standardised activities [14]. While the remaining papers all included a more ecologically valid environment (such as a play centre, the home, or a childcare centre), they mostly used a structured set of activities rather than free play. Only two investigations were cross-validated in a completely uncontrolled free-living space [33,34]. Collectively, these studies found that the accuracy of the posture and movement prediction found in a laboratory-based study was reduced by 15–20% when assessed in free-living conditions [33,34]. All studies used either human coding of video or direct observation as the gold standard. While this gold standard was demonstrated to be sufficiently accurate and reliable [7], it might limit the study design to controlled settings, given that videoing and observing free-living conditions is time-intensive and impractical for longer durations. Therefore, while future work should include a cross-validation component in free-living environments, this would be facilitated by computational approaches, such as machine learning, in the processing of gold standard data.

### 4.4. Hardware

There was very little consensus on the type of hardware applied and how it was used across the studies reviewed. Actigraph was the most frequently used commercial device (eight studies), with a diverse range of other commercial and non-commercial sensors otherwise applied. It is generally accepted that more sensors will increase the accuracy of the algorithm developed [41]. However, this is not practical when long-term or large-scale monitoring is planned. Given that most studies aimed to develop a system that could be used for multiple day recording in large samples to establish links between postures and movements with childhood health, they also generally only used a single sensor. There was diversity in the location of single sensors, although the hip and the wrist were the most common locations. One paper compared the accuracy between locations and found that a hip sensor was slightly more accurate than the wrist for identifying 23 different activities [34]. Research in healthy adults has demonstrated that a single sensor located on the thigh is more suitable for differentiating sitting and standing in adults [10]; however this has not yet been confirmed in children. While it appears that there is some consensus that a single sensor is optimal for long-term childhood activity tracking, more research is required to determine which location is optimal.

### 4.5. Software

The results of this review highlight a range of different software prediction approaches used to objectively record children’s postures and movements. The one consistency was that machine learning methods were favoured. Only three studies used algorithms that required human-specified criteria, suggesting that this approach is becoming less favoured amongst the research community. Although the majority of studies employed conventional machine learning, the most recent papers have started utilising deep learning, consistent with human activity recognition (HAR) work in adults. In the conventional machine learning models, there was considerable diversity relating to feature extraction. There was no consistency in choice of overlapping or non-overlapping windows, the size of windows, or the features selected. Some papers highlighted the influence of these aspects on accuracy, with longer window sizes appearing more suitable. There was also a large range of traditional machine learning models used but with Random Forests being the most common. However, the interplay between features, window size, and model makes recommendations on optimal approaches difficult.

### 4.6. Validation Approach and Accuracy

Traditionally, machine learning models are validated by splitting the data into training, validation, and test data sets and, in the case of HAR, by splitting by participant rather than by random windows of data. The training and validation data sets are then used to train the model, with the validation set used to tune the hyperparameters of models coming from the training set. Then, the accuracy is defined by applying the model to the previously unseen test data. More recently, validation has been performed through cross-validation techniques such as n-fold and leave-one subject-out (LOSO) cross-validation. In these scenarios, the test data set is not seen at all in the model development, and this can be considered to give independent accuracy measures. However, this approach has not been undertaken in any of the papers reviewed (with the exception of Ahmadi, where a previous model derived from laboratory data was tested against free-living data [33]). Further, this practice requires large, labelled datasets which are not readily available for children. One potential solution is for research groups to share data where similar accelerometers in similar positions have been utilised, so one group can test their model on unseen data from another group and vice versa. Many papers in this review have employed n-fold and LOSO cross-validation methodologies but have applied them across all of the data collected.

The results of this study highlight a very large range of prediction accuracies across each of the examined postures and movements. Walking and running were most commonly examined, with accuracy ranging from as low as 9% (walking) to as high as 100% (running). Importantly, for the 13 postures and movements evaluated in samples of more than 25 children (with the exception of ‘pivoting’), an accuracy of greater than 80% was reported. This is meaningful, as this 80% threshold of accuracy has been largely accepted as the cut-off for acceptable implementation of a model. However, given that only one study assessed their model in free-living conditions, caution should still be applied when interpreting these results.

### 4.7. Strengths and Weaknesses

A strength of this review is that the author team included both human health and computational expertise, enabling transdisciplinary understanding and translation of the findings. A further strength was that the review included quality assessment of the papers, which is not always included in machine learning systematic reviews, again likely reflective of transdisciplinary differences. Lack of quality assessments in other machine learning reviews may also have been due to the limited applicability of most quality assessment tools for machine learning-type studies, reflected in the need to modify the COSMIN quality assessment to ensure it met the needs of this study.

A limitation of the study was that it did not include studies of older children which may have useful information, but these studies were partly covered by another recent review [14]. This review was focused on studies that utilised machine learning and algorithm-based approaches to wearable movement sensors. Thus, papers that objectively measured movements and postures using other data sources, such as video data, were not summarised.

### 4.8. Implications

This review suggests a number of implications for future machine learning model developments, including the importance of ensuring an adequate sample in terms of size and representativeness, sample age and sex, sensor location, separation of training and testing data, laboratory and field testing, and inclusion of a broad range of postures and movements commonly used by children. Further research should also explore the strengths and weaknesses of various machine learning approaches.

## 5. Conclusions

Young children’s postures and movements are critical to their current and future health and development, so high-quality evidence from robust measures is essential to understanding how to support healthy development. The findings of this review suggest that the rapidly developing machine learning field has demonstrated there is potential to substantially enhance the quality of such evidence.

## Figures and Tables

**Figure 1 sensors-23-09661-f001:**
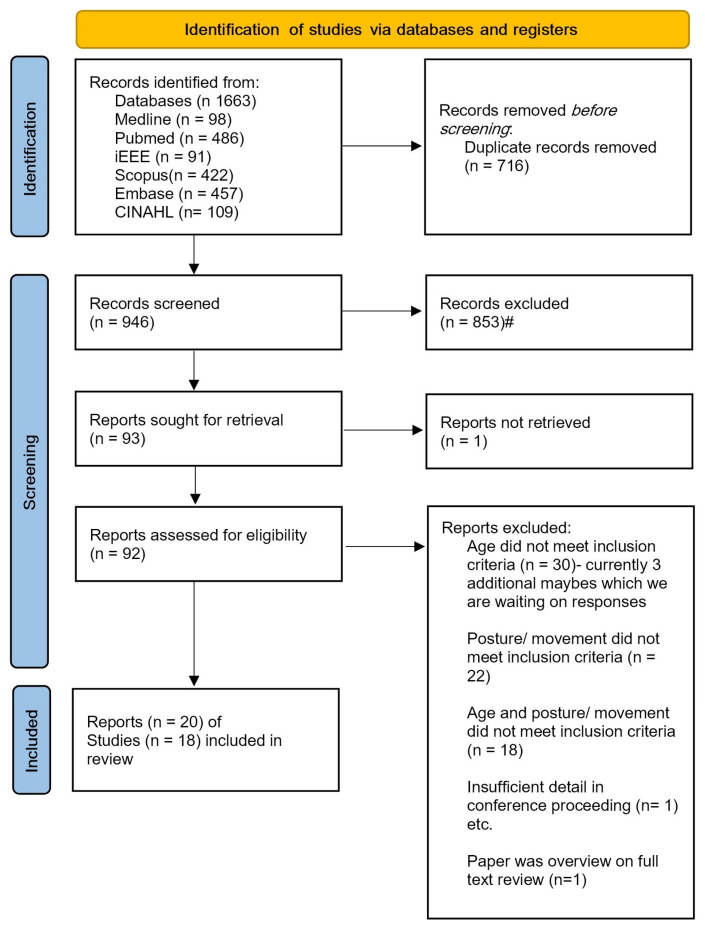
PRISMA flowchart of included studies. #Screening took place using “Research Screener” artificial intelligence screening software. Nine rounds of screening (50 in each round) took place before the screeners determined that no further useful papers were being shown.

## Data Availability

No data were created for this review.

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
