# Peer review of "Objective Measurement of Posture and Movement in Young Children Using Wearable Sensors and Customised Mathematical Approaches: A Systematic Review"

_sensors, 2023, doi:10.3390/s23249661_

Round 1

Reviewer 1 Report

Comments and Suggestions for Authors

The paper follows the structure of a systematic review, is well written but some information could be better reported and detailed.

Some minor aspects should be addressed and are listed below in detail-

The Quality assessment developed with the Cosmin guidelines had to be recalled and reported briefly to better understand the reader the analysis carried out by the researchers on the selected articles.

It is not clear why the researchers included 6 criteria for the analysis of the "Validity criterion", while in the guidelines it is based on 5 aspects as also reported in the document (lines 195-198). The choice to combine criteria 4 with 5 should be better described given that for criterion number 4 the parameters to be verified in the selected articles are not reported.

To improve the reading of paragraph 3.2, I would recommend dividing the paragraph into subsections, each specific for the fields shown in Table 3. Therefore a section for the selection of Participants, one for the Purpose of the Study and so on.

From reading the details in Table 3 about the study design, it is not clear why Parkka's 2010 article was included in the review given that the patients studied were a sample of the population aged 4 to 37 years where however only a 4 year old child was present. Wasn't the purpose of the review to analyze the behavior of young people?

Reviewer 2 Report

Comments and Suggestions for Authors

This review paper investigates researches regarding the objective measurement of children's posture and movement using wearable sensors. This paper was considered to make some contribution in this research area. However, a number of problems were found, and it was considered that a major revision is necessary to publish.

The most significant problem is the discrepancy between the title and the content. The term "machine learning" is used in the title, but the papers reviewed in this paper are not limited to it. As for "accelerometry", it also includes papers that use a wider range of sensors. The authors need to carefully consider how these should be expressed in the title.

Machine learning is now getting more attention. However, there are fewer descriptions for machine learning in this papers, and it may not meet readers expectations. For example, there is no aggregate results for machine learning methods, nor discussion for them. Section 3.3 and 4.5 need to be strengthened.

I failed to understand the procedure of identification shown in Figure 1. It also contains calculation errors. Please compare it with the figure as in reference [14].

I thought the structure of chapter 3 (Result) should to be improved. From the beginning to 3.1 are the results of the analysis done by authors, while 3.2 and after are characteristics  of the included papers. These might be split up for readability ?

In the all Tables, please add reference number for each paper. It was too hard to track the included papers.

The description of the items used in Table 1 and 2 was not sufficiently explained in the body of the document. It was difficult to read what was altered and how it was decided to be this way. Furthermore, there were several calculation errors in table 1 and 2.

Table 3 to 5 was considered to be the core of this study, but they were very difficult to read and gave the impression that they were not sufficiently organized. I think that the tables need to be revised significantly with reference to other review papers.

In chapter 4 (Discussion), it was difficult to read the relationship between the results described in previous chapter and the author's assertions, giving the impression that the assertions are begin made abruptly and subjectively.

For these reasons, I would like to recommend a major revision for authors.

Specific Comments:

L33: All keywords overlapped with title. I thought it is not appropriate.

L120: "additional file A" must be "Appendix A".

L122: "since January 2012" but two papers included the papter were published before 2012.

L133-L137: It was difficult to follow and needs rephrasing.

L139: Is "Endnote" nesessary to state ?

L149: Is it correct to refer [18] in this context ?

L150-152: There is a lack of explanation of "items" and it is hard to understand what the authors did for the assessment.

L179: "three papers" are refered but only [19,20] are mentioned ?

3.2, 3.3, 3.4,: The descriptions of the included papers are not standardized, i.e. "[26,27]", (n=2[21,28]), (n=6).

L304-306: Several "and"s makes the sentence unclear.

Table6: It is hard to read the table because there are no description in the body of document. The first column is also hard to understand.

L340: I could not find "28%" and "0%" in Table 6.

L367: "over 30 activities" is descrived in chapter 3 ? 

L368: Authors need to show why it is "urgent".

L370: It is unclear why paper [14] is refered here.

L407: More explanation is needed for this assertion.

L426: What is "computational approaches" ? The intention of this sentence is unclear.

L430: Is "locaiton" of attached sensor "Hardware" ?

L449: It was hard to agree that the approach was "less desirable". Please strengthen the discussion.

L453: "The traditional machine learning models" is not defined.

L470,L472: Two consecutive "However".

L499-502: Please clarify how this relates to your research aim.

L510: "generalisability, complexity and burden" are discussed in this paper ?

L510: "Finally, ..." It was unclear that on what basis/discussion did you make this claim.

L520: "However..." It was unclear that on what basis/discussion did you make this claim.

Round 2

Reviewer 2 Report

Comments and Suggestions for Authors

The authors had addressed most of my concerns.

I would recommend it for acceptance after the minor points listed below are double checked by authors.

In Figure 1, does asterisk symbols mean something ?

In Table 1, the number of good or adequate for "Trost, 2018" might be 8 ? Number of studies scored G or A for "General design items" 5 and 9 is correct ?

In Table 2, "Number of good or adequate" seems to be wrong for Suzuki, Li and Jun.

I thought that if Table 1 contains a reference number in the labels like "Parkka, 2010 [21]", it may help readers to find the reference paper with phrases in the text such as "from 1 [21] to 100 [22] participants".
